# Exposure to Endocrine Disruptors (Di(2-Ethylhexyl)phthalate (DEHP) and Bisphenol A (BPA)) in Women from Different Residing Areas in Italy: Data from the LIFE PERSUADED Project

**DOI:** 10.3390/ijms232416012

**Published:** 2022-12-16

**Authors:** Fabrizia Carli, Sabrina Tait, Luca Busani, Demetrio Ciociaro, Veronica Della Latta, Anna Paola Pala, Annalisa Deodati, Andrea Raffaelli, Filippo Pratesi, Raffaele Conte, Francesca Maranghi, Roberta Tassinari, Enrica Fabbrizi, Giacomo Toffol, Stefano Cianfarani, Cinzia La Rocca, Amalia Gastaldelli

**Affiliations:** 1Institute of Clinical Physiology, National Research Council, 56124 Pisa, Italy; 2Centre for Gender-Specific Medicine, Istituto Superiore di Sanità, 00161 Rome, Italy; 3Dipartimento Pediatrico, Universitario Ospedaliero “Bambino Gesù” Children’s Hospital, 00165 Rome, Italy; 4Department of Systems Medicine, University of Rome Tor Vergata, 00133 Rome, Italy; 5Unità Operativa Dipartimentale di Pediatria, Asur Marche Area Vasta 3, Ospedale di Civitanova Marche, 62012 (MC), Italy; 6Associazione Culturale Pediatri, 09070 Narbolia, Italy; 7Department of Women’s and Children’s Health, Karolinska Institute and University Hospital, 171 77 Stockholm, Sweden

**Keywords:** di-(2-ethylhexyl) phthalate, bisphenol A, endocrine-disrupting chemicals, women exposure, geographical area, human biomonitoring

## Abstract

Phthalates and bisphenol A (BPA) are plasticizers used in many industrial products that can act as endocrine disruptors and lead to metabolic diseases. During the LIFE PERSUADED project, we measured the urinary concentrations of BPA and Di(2-ethylhexyl)phthalate (DEHP) metabolites in 900 Italian women representative of the Italian female adult population (living in the north, centre, and south of Italy in both rural and urban areas). The whole cohort was exposed to DEHP and BPA with measurable levels above limit of detection in more than 99% and 95% of the samples, respectively. The exposure patterns differed for the two chemicals in the three macro-areas with the highest urinary levels for DEHP in south compared to central and northern Italy and for BPA in northern compared to central and southern Italy. BPA levels were higher in women living in urban areas, whereas no difference between areas was observed for DEHP. The estimated daily intake of BPA was 0.11 μg/kg per day, about 36-fold below the current temporary tolerable daily intake of 4 μg/kg per day established by the EFSA in 2015. The analysis of cumulative exposure showed a positive correlation between DEHP and BPA. Further, the reduction of exposure to DEHP and BPA, through specific legislative measures, is necessary to limit the harmfulness of these substances.

## 1. Introduction

Phthalates and bisphenols have been manufactured in large quantities by industry since the early 1900s as additives to make plastic polymers variably rigid, transparent, elastic, soft, or resistant [1,2]. For these properties, phthalates and bisphenols, of which di 2-ethylhexyl-phthalate (DEHP) and bisphenol A (BPA) are the most common, have a wide use and are found in many industrial products such as toys, medical devices, and plastic bottles but also in personal care products, cosmetics, perfumes, and food packaging [3]. Food contact materials represent the 20% of total exposure to BPA, with canned food as the main source of external dietary exposure and thermal paper the second [4]. In 2016, the Committee for Risk Assessment (RAC) of the European Chemicals Agency published a new regulation (Commission Regulation 2016/2235 of 12 December 2016) regarding the Registration, Evaluation, Authorization, and Restriction of Chemicals (REACH), which stated that BPA “shall not be placed on the market in thermal paper in a concentration equal to or greater than 0.02% by weight after 2 January 2020”. In 2018, following the opinion published by EFSA, the specific migration limit of BPA for plastic materials and articles was decreased from 0.6 mg/kg of food to 0.05 mg/kg of food (Commission Regulation (EU) 2018/213 of 12 February 2018).

Regarding DEHP, the European Food Safety Authority (EFSA) in 2005 and then in 2019 reported that food is the main source of exposure [5].

However, the real exposure to these chemicals is greatly unknown because of the limited number of studies performed up to now or the limited number of subjects enrolled in the studies. This is particularly true for Italy. Thus, the large biomonitoring (HBM) study LIFE PERSUADED was designed and performed from 2015 to 2017 to establish the exposure to phthalates and BPA in the Italian population throughout the territory and to provide background levels in mothers and their children. To accomplish this, 900 mother–child pairs (boys and girls with age 4–6 years, 7–10, and 11–14 years) were enrolled in three different geographical macro-areas, i.e., the north, centre, and south of Italy, from urban and rural areas. All subjects filled out dedicated questionnaires and provided a urine sample for the measurement, by high sensitivity mass spectrometry, of the levels of BPA and DEHP metabolites (MEHP, MEOHP, and MEHHP). More details on this project have been previously reported [6].

DEHP and BPA are included in the list of substances subjected to authorization under the REACH regulation and candidates for substitution, as they are of very high concern (ECHA/PR/18/01) (ECHA/ED/108/2014); both compounds are recognized as endocrine-disrupting chemicals (EDCs) with effects on the endocrine and reproductive systems, thyroid, liver, kidneys, and nervous system and cancer development [7,8,9,10,11]. Further, both DEHP and BPA are considered “obesogenic” EDCs due to the associations of their exposure with metabolic syndrome, obesity, type 2 diabetes mellitus, and hepatic fat accumulation [12,13]. Due to the recent evidence on adverse effects of BPA on the immunological system, the EFSA and the Food and Drug Administration (FDA) are re-evaluating the tolerable daily intake [14,15], suggesting to lower the value from the current 4 μg/kg day to 0.04 ng/Kg day [14].

The metabolism of DEHP and BPA after ingestion is rapid, and it occurs mainly in the liver, where DEHP is hydrolyzed to mono(2-ethylhexyl) phthalate (MEHP) by unspecific lipases [16]; then, MEHP is oxidized to mono(2-ethyl-5-hydroxyhexyl) phthalate (MEHHP) and to mono(2-ethyl-5-oxohexyl) phthalate (MEOHP). Monoesters and the oxidative metabolites are conjugated to form hydrophilic glucuronide conjugate metabolites, which are easily excreted in urines [16,17,18,19]. The major metabolic pathway of BPA in humans is the BPA-glucuronidation by the enzyme UDP-glucuronyl although BPA can also be conjugated via sulfation by sulfotransferases [20], which are the major forms present in the urines. In addition, in vivo and in vitro studies suggest that BPA may be subjected to oxidation with the production of metabolites, among which 4-methyl-2,4-bis(p-hydroxyphenyl)pent-1-ene (MBP) was found to increase estrogenic activity [21] as well as to trigger endoplasmic reticulum (ER) stress [22].

In this paper, we report the HBM results of the LIFE PERSUADED study for mothers with comparison of exposure levels according to living area, which completes the data already published for the children [23,24]. Results of this large HBM study may support the adoption of legislative measures aimed at limiting the exposure to DEHP and BPA.

## 2. Results

### 2.1. Characteristics of the Cohort

The clinical characteristics of the mothers participating to the study are shown in Table 1. The median age was 41 years ranging from 24 to 67 years. Mothers from southern Italy were younger compared to those living in central Italy, and mothers living in urban areas were older than those in rural areas. Although the great majority of these women was lean (median BMI of 22.3 kg/m^2^; 20.3–24.8 interquartile range), BMI data showed a significant trend to higher values from the north to the south of Italy, while no significant difference was observed between mothers residing in rural or urban areas.

### 2.2. Phthalates and BPA Levels in Italian Women

In this cohort of Italian women, DEHP was detected in over 99% of the subjects (Table 2): MEHP values above LOD were detected in about 99% of the population and the secondary metabolites MEHHP and MEOHP in about 97% and 99% of the population, respectively. Detectable BPA levels (free and glucuronide forms) were found in 95.65% of the 898 women included in this study. Table 2 reports unadjusted urine levels in μg/L and levels adjusted to creatinine content, in μg/g, for each DEHP metabolite and their sum as well as for BPA. As expected, the secondary DEHP metabolite MEHHP was found at higher levels compared to the other metabolites.

### 2.3. Phthalates and BPA Levels by Residing Area

Significant differences were found among women residing in the different macro-areas of Italy. MEHP, MEHHP, MEOHP, and sum of the DEHP metabolites levels were significantly higher in women living in southern compared to those living in northern and central Italy for both unadjusted and adjusted urinary concentrations (except adjusted MEHP concentrations). Figure 1A shows the distribution of DEHP metabolites adjusted to creatinine content (µg/g) in the north, centre, and south of Italy, whereas Table 3 reports data of the sum of DEHP metabolites in each macro-area and area. (Single DEHP metabolites data are reported in Appendix A.)

No significant differences were observed between women residing in urban or rural areas for the sum of DEHP metabolites (Table 3) and the single metabolite levels (Appendix A).

Significantly higher BPA-unadjusted and -adjusted levels were found in women residing in northern Italy compared to those residing in the other two macro-areas (Table 3). Further, BPA-unadjusted and -adjusted levels were higher in women residing in urban than in rural areas (Figure 1B).

Data stratified by areas in each macro-area are reported in Table 4 for the sum of the DEHP metabolites and BPA and in Appendix A for the single metabolites. Significant differences between residing areas were found for MEHP (both as µg/L and µg/g crea) in southern Italy, being higher in rural than urban areas (Appendix A), and for the sum of DEHP metabolites as adjusted values in central Italy, being higher in urban than rural areas (Table 4).

For the sum of DEHP metabolites as well as for the single MEHP, MEHHP, and MEOHP, higher unadjusted and creatinine-adjusted levels were observed in rural areas of southern Italy compared to the north and centre. Further, MEOHP-adjusted levels were higher in rural areas of northern then of central Italy. (Table 4 and Appendix A). In urban areas of southern Italy, women had higher unadjusted MEHHP levels compared to the north and centre and adjusted MEHHP levels compared to the north; in addition, unadjusted and adjusted MEOHP levels were higher in urban areas of the south compared to the north and centre (Appendix A), and there was a higher unadjusted sum of DEHP metabolite levels compared to the other two areas, whereas creatinine-adjusted levels were higher only compared to central Italy (Table 4).

In central and southern Italy, BPA levels in women residing in urban areas were higher than in women from rural areas, both as unadjusted and adjusted values (Figure 1B and Table 4). Women from rural areas of northern Italy had higher BPA-unadjusted and -adjusted levels compared to women from the other two rural areas. Further, urban women from northern Italy had higher BPA-adjusted levels compared to urban women from the south of Italy.

### 2.4. Phthalate Metabolic Rates

The analysis of the relative metabolic rates of DEHP metabolites and their molar percentages evidenced that women residing in southern Italy had a higher metabolic conversion rate from MEHP to MEHHP + MEOHP (RMR1) compared to women from the other two macro-areas (Table 5). Accordingly, MEHP molar percentage in these women was significantly lower.

The rate of the second metabolic conversion (RMR2) was significantly different among women of the three macro-areas, with increasing values from the centre to the north and to the south of Italy. As a consequence, the same pattern was observed for the MEOHP molar percentage. No difference in RMR1 and RMR2 were observed between women residing in rural or urban areas.

Analysing metabolic rates and molar percentages by age categories, women aged 30–50 years had higher RMR1 and lower %MEHP values compared to women >50 years; conversely, RMR2 significantly decreased with age, with higher values in women aged 20–30 years (Appendix A).

No significant difference in RMR1 and %MEHP values were observed in women according to different BMI categories. Otherwise, significantly higher RMR2 values were observed in women with a BMI > 30 compared to women with BMI < 25 (Appendix A). Accordingly, women with BMI >30 had higher %MEOHP values compared to women with BMI < 25 but also with BMI in the overweight range (25–30). Women with BMI > 30 had also lower %MEHHP values compared to overweight women.

### 2.5. Daily Intake (DI) of BPA and DEHP

The calculated daily intake of BPA for total enrolled women providing their BW (N = 879) was 0.11 µg/kg bw per day as geometric mean (0.11–0.12 95% CI) (Table 6). Considering the current tolerable daily intake (TDI) of 4 µg/kg bw per day [4], only one woman exceeded this value, and 2.28% women (N = 20) had a daily intake between 1 and 4 µg/kg bw per day.

The estimated daily intake of BPA was significantly higher in the mothers living in northern Italy and in urban areas, especially in the centre and south of Italy. In particular, women living in rural areas from northern Italy had higher daily intake values than women in rural areas from the centre and south of Italy. Furthermore, urban northern women had higher values than urban women from the south. In addition, women aged 30–50 years had higher daily intake values than women aged > 50 years.

The calculated daily intake of DEHP for total enrolled women providing their age, BW, and height (N = 881) was 4.86 µg/kg bw per day as geometric mean (4.66–5.08 95% CI) (Table 7). The daily intake of women living in the rural south was significantly higher than in the north and centre, while in the urban area, DI in the south was only higher than in the north. No significant difference was found within each macro-area of the DI of women living in rural or urban areas.

### 2.6. Analysis of Correlations

The sum of DEHP metabolites and BPA levels were positively correlated in the total population of Italian women, in central Italy, and in urban areas both as unadjusted and adjusted values (Appendix A). A positive correlation was found only for unadjusted values in southern Italy and for adjusted values for women living in rural areas.

Only for the sum of DEHP metabolite did we find a positive correlation with age as adjusted levels both in the total population and in northern Italy although with very low rho coefficient (Appendix A). A positive correlation was also observed in women from southern Italy between the adjusted sum of DEHP metabolite levels and BMI (Appendix A).

### 2.7. Reference Values in Women Population

Reference values (RV95) in the total Italian women population were 101 and 30.6 µg/L for the sum of DEHP metabolites and BPA, respectively (Table 8). RV95 values for ΣDEHP were higher than the upper bound for the total population in the south of Italy, especially the urban areas, and in younger mothers (age range 20–30 years). BPA RV95 values exceeded the upper bound for the general women population in the north of Italy and in rural areas, in particular the northern ones.

## 3. Discussion

The LIFE PERSUADED project is a large HBM study that enrolled 900 mother–child pairs from 2015 to 2017 with the aim to establish the level of co-exposure to phthalates and BPA in the Italian population [6], where such data were still lacking. The strength of the study is that it is the first determination of the background and reference values in Italian women, is useful in risk assessment, and provides the evaluation of the co-exposure to DEHP and BPA. Moreover, the large number of mother–child pairs enrolled and the corresponding collected data during the LIFE PERSUADED project allow an accurate calculation of all these values also as a function of the different macro-areas and areas of residence.

The mothers that participated to this study had a median age of 41 years (38–44 interquartile range), and they were relatively lean (median BMI of 22.3 kg/m^2^; 20.3–24.8 interquartile range), with a balanced distribution among macro-areas and urban vs. rural areas. However, women living in southern Italy and in rural areas were slightly younger and more overweight compared to mothers living in the north and centre of Italy or in urban areas and showed a positive correlation between DEHP levels and BMI. In addition, in the whole population and specifically in the north of Italy, a positive correlation was observed between DEHP metabolites and age, which is in contrast to what was observed in children, for whom higher values were observed in the youngest [24].

The high detection of DEHP metabolites (99%) and BPA (>95%) indicates high exposure to these chemicals, as also previously shown in other European countries [25,26,27,28,29,30].

DEHP exposure in Italian mothers is within the range observed by European HBM studies in women, e.g., the DEMOCOPHES project [30]. Instead, BPA exposure was 5.79 μg/L, i.e., about three times higher than in other countries, as reported in a recent meta-analysis including 28,353 participants from different parts of the world, with levels ranging from 0.81 to 3.50 μg/L [31], but similar to another study conducted in Italy in the Tuscany Chianti area, involving 715 adults between 20 and 74 years old, where GM of BPA was 5.14 μg/L [32]. Since similar BPA levels were found in their children [23], it can be hypothesized that the Italian population is more exposed to BPA as a possible consequence of lifestyles, eating habits, or environmental contamination.

With respect to geographical areas, DEHP exposure was higher in southern than in central and northern Italy, with mothers’ exposure similar to that of their children [24]. On the contrary, for BPA, mothers and children showed different exposure patterns [23]; while the highest BPA exposure was recorded for mothers living in northern Italy, for children, the highest levels were recorded in the south. Therefore, common determinants of exposure for mother–child pairs based on area of residence are shown for DEHP, thus being independent of age and lifestyle; conversely, different determinants of exposure for young and adult populations may have resulted in diversified exposure to BPA.

No difference was observed among women living in urban vs. rural areas in DEHP metabolites, similar to their children [24] and to other European countries [25,26,27,28] with the exception of Ireland, where MEOHP levels were found to be higher in women residing in the urban area [28]. BPA exposure was instead higher in Italian mothers living in urban areas, with a trend similar to what was observed in their children [23]. This is probably due to different habits between differently urbanized areas that should be carefully considered in the evaluation of the risk.

The concentrations of DEHP and BPA were positively correlated, in particular in central and southern Italy, suggesting possible common sources or an extensive use of plastics, resulting in a consequent cumulative exposure. This evidence is particularly relevant considering the effects exerted by the mixture on the glucose metabolism, thyroid, and cancer development exerted by the mixture and shown in in vivo studies [9,33,34,35]. In addition, we have previously shown that the combined exposure of juvenile rats to DEHP and BPA, at the same doses recorded in this HBM study, was associated with synergic action in the metabolic system and antagonism action in the reproductive and endocrine systems [36]. Thus, the evidence gathered in the present HBM study further supports the concerns about possible adverse effects exerted by these plasticizers on young and adult populations, hopefully serving as a basis for policy regulators to introduce greater restrictions on their use.

We also collected data on lifestyle and food consumption habits, as determinants of exposure, through a dedicated questionnaire and food diary (unpublished data). Preliminary results showed that higher levels of phthalates and BPA are associated with the use of single-use plastics. We evaluated the BPA daily intake to verify a possible exceedance of the tolerable daily intake (TDI) of 4 µg/kg per day established by EFSA in 2015 [4], while the DEHP daily intake was 4.68 µg/kg per day, i.e., within the EFSA limit of 50 µg/kg per day, in line with other European studies [29]. BPA daily intake in Italian women was 0.11 µg/kg per day, thus within the current TDI. It is of note that EFSA and FDA are currently re-evaluating the TDI [14,15], proposing to reduce the value to 0.04 ng/kg per day [14] due to collected evidence on adverse effects of BPA on the immune system, possibly representing the most sensitive target. In this scenario, all subjects enrolled in this HBM study would have TDI well above the new proposed limit, i.e., 2750 and 4250 times higher for the mothers and their children, respectively. This observation raises concerns about BPA exposure in general, particularly in Italy, where internal levels are higher compared to other European and non-European countries, making it urgent to adopt measures to reduce the exposure to this plasticizer (and thus reduce the risk of adverse health outcomes).

The relative metabolic rates of DEHP metabolites in the Italian macro-areas showed that women living in southern Italy had a higher metabolic conversion of MEHP into the secondary metabolites (MEHHP + MEOHP), with consequent lower percentages of molar MEHP, especially in women aged 30–50 years. Moreover, RMR2, which reflects the oxidation of MEHHP to MEOHP, was significantly different among women of the three macro-areas, with a pattern of south > north > centre of Italy. Thus, in women, both metabolic conversions differed across macro-areas, whereas in children, only RMR2 displayed differences but with a pattern of north > centre > south [24]. Interestingly, an RMR1 pattern similar to women, decreasing from south to north, was observed in older children; hence, as previously hypothesized, RMR1 competence seems to increase with age, while RMR2 mostly varies according to the geographic area of residence [24]. Although higher RMR2 was found in women aged 20–30 years, these data should be evaluated with caution because the number of subjects in the group is low; indeed, all other women > 30 years had comparable RMR2 values, thus supporting the previous hypothesis. Overall, the observed differences may be explained by considering that these metabolic conversions occur in the intestine and liver by enzymes of the cytochrome P450 chain, which have numerous polymorphisms within and among populations [37]; the high genetic variability of the Italian population [38] might explain, at least in part, these results.

RV95 values, which are useful parameters for the identification of the highest exposure scenario, were defined for both DEHP (∑MEHP, MEHHP, MEOHP) and BPA in Italian women aged from 24 to 67 years, equal to 101 µg/L and 30.6 µg/L, respectively. Compared to their children, RV95 for DEHP was higher in children (168 µg/L), whereas for BPA, children had a lower value compared to mothers (24.1 µg/L). Furthermore, the large sample size made it possible to derive RV95 values for subgroups of the female population based on geographical areas and age groups, making possible a more accurate assessment of exposure and risk.

## 4. Materials and Methods

### 4.1. Study Population

In the LIFE PERSUADED project, a HBM study was performed enrolling 900 Italian healthy mother–child pairs between 2015 and 2017, equally distributed in northern, central, or southern Italy [6].

Subjects were also equally selected who resided in urban or rural areas, as defined by population density (> or <150 inhabitants/km^2^) and number of inhabitants (> or <50,000 inhabitants) on the basis of the *Italian Institute of Statistics* (ISTAT) database. Included subjects had to be at their residence for at least 6 months prior the enrolment. Mothers were also equally distributed according to their children sex and age category (4–6 y, 7–10 y, 11–14 y). The enrolment was carried out in different Italian regions by family paediatricians of the Italian Health System, who voluntarily joined the HBM study through the Associazione Culturale Pediatri (ACP, https://www.acp.it, accessed on 30 November 2022) and the Federazione Italiana Medici Pediatri Marche (FIMPM, http://www.fimpmarche.it, accessed on 30 November 2022) networks, as previously described [6].

Inclusion criteria were healthy status, being a mother of children between 4 and 14 years, and with a body mass index between 5th and 85th percentile. Mothers in gestational status or in breastfeeding were excluded.

The whole study was approved by the Ethical Committee of the Istituto Superiore di Sanità. The enrolled mothers signed an informed consent to provide a first-morning urine sample for the measurement of DEHP and BPA exposure and filled out a questionnaire with information on residential environment, socio-demographic, and lifestyle data as well as on food habits. Participants were assigned an alphanumeric code to guarantee anonymity [6].

Subjects were excluded from final analysis if urine sample was not available. Final analysis was performed on 898 mothers equally distributed in northern (N = 300: n = 150 rural and n = 150 urban), central (N = 299: n = 149 rural and n = 150 urban), and southern (N = 299: n = 149 rural and n = 150 urban) areas. BMI data were available for 879 women, whereas only 655 women declared their age in the questionnaires.

### 4.2. Sample Collection and Measurements of Phthalates and BPA Levels

The concentration of DEHP metabolites and BPA was measured in first-morning urine samples, as previously described [23,24]. Concentrations of DEHP metabolites (e.g., MEHP, MEHHP, and MEOHP) and BPA were determined using internal standards as described in detail in [39]. Limit of detection and quantification (LOD and LOQ) were for MEHP 0.28 and 0.58 ng/mL, for MEOHP 0.18 and 0.48 ng/mL, for MEHHP 0.11 and 0.24 ng/mL, and for BPA 0.157 ng/mL and 0.523 ng/mL, respectively. Both analytical methods were validated in a proficiency test (ICI/EQUAS) in the frame of the HBM4EU Project (https://www.hbm4eu.eu/online-library/, accessed on 30 November 2022), as previously reported [39,40].

Data were then normalized to urinary creatinine concentrations as determined by the Jaffe’s method (Beckman, Brea, CA, USA), all in the range 0.3–3 g/L.

### 4.3. Relative Metabolic Rates and Percentage Fractions and Reference Values

Relative metabolic rates (RMR) and percentage fractions of DEHP metabolites in molar concentration were calculated as previously described [24,41]:RMR1 = ([MEHHP] + [MEOHP])/[MEHP]
representing the metabolic conversion of MEHP into the secondary metabolites (MEHHP + MEOHP) and

RMR2 = ([MEOHP]/[MEHHP]) × 10
representing the rate of oxidation from MEHHP to MEOHP.

Percentage fractions of each DEHP metabolite were calculated with respect to the sum of all metabolites.

Reference values (RV95) for both DEHP as sum of its metabolites and BPA were calculated as 95th percentile with corresponding 95% confidence intervals, as previously described [24]. RV95 was calculated for total population, macro-areas, areas, and age class.

### 4.4. Daily Intake of BPA and DEHP

Daily intake of BPA was calculated by using the EFSA equation based on BPA concentration in urine volume [4]:Intake BPA (µg/kg per day) = C_BPA_ × V_urine_/BW
where C_BPA_ is the BPA concentration in urine sample (µg/L), and V_urine_ is the daily urinary output rate, assumed to be 1.2 L/day. Among the 898 eligible enrolled women, 19 did not provide their BW; thus, the calculation was performed on 879 women.

Daily intake of DEHP was estimated as sum of DIs of each DEHP metabolites, calculated as method reported by Koch et al. [18,42]:

Intake DEHP (µg/kg per day) = (UE(µg/g crea) × CE(mg/kg/day))/(F_UE_ × 1000 mg/g) × (MW_DEHP_/MW_metabolite_)

where

UE = urinary excretion of DEHP metabolites (µg/g crea);

CE = creatinine excretion, set to be 18 mg/kg/day [42];

F_UE_ = urinary excretion fraction according to [18,42].

MW = molar weights of DEHP and corresponding metabolites

Among the 898 eligible enrolled women, DI of DEHP was performed on 881 women.

### 4.5. Statistical Analysis

Statistical analysis was performed with STATA 14.2 (StataCorp, 4905 Lakeway Drive, College Station, TX, USA), setting significance at *p* < 0.05 for all the statistical tests performed.

Since data are not normally distributed, non-parametric tests were performed to evaluate statistical differences among different groups. For two-levels strata (e.g., area), the Mann–Whitney test was performed, whereas for three-levels strata (e.g., macro-area), the Kruskal–Wallis test was performed with Dunn’s post hoc evaluation where applicable. Differences between groups were calculated for both unadjusted and creatinine-adjusted urinary levels.

Correlation analysis was performed by Spearman’s test with Bonferroni correction.

For additional analyses, age and BMI were categorized as follows: age (20–30, 30–40, 40–50, >50) and BMI (<25, 25–30, >30).

Analytical data on DEHP metabolites and BPA in the Italian women involved in this study will be made available on the Information Platform for Chemical Monitoring of the European Commission (https://ipchem.jrc.ec.europa.eu,), as already done for data on children (https://ipchem.jrc.ec.europa.eu/index.html#showmetadata/LIFEPERSUADED; https://ipchem.jrc.ec.europa.eu/), accessed on 30 November 2022.

## 5. Conclusions

The data obtained from the LIFE PERSUADED project indicate ubiquitous exposure to DEHP and BPA for the Italian adult women. The results obtained in Italian mothers confirm the results in children, indicating women living in southern Italy as the most exposed to phthalates and with less ability to process the secondary metabolite MEHHP. Conversely, women living in northern Italy and urban areas were more exposed to BPA. Overall, the area of residence is confirmed as an important determinant of exposure to plasticizers. Therefore, data on background levels, RV95, and dietary intake of DEHP and BPA in Italian mothers provide a contribution to the assessment of the risk associated with the exposure to plasticizers in adults in relation to the geographical area. Considering the health risk associated with exposure to plasticizers, these data should guide the implementation of policies aimed at reducing the public health impact of these chemicals and improving communication strategies to increase women’s awareness for protecting their health by reducing exposure to plasticizers.

## Figures and Tables

**Figure 1 ijms-23-16012-f001:**
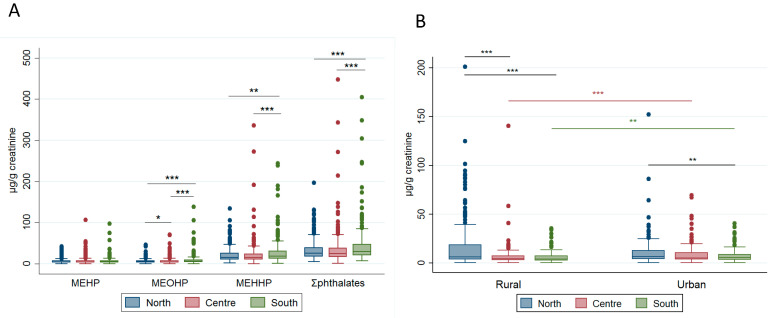
Box plots for DEHP metabolites (panel (**A**)) and BPA levels (panel (**B**)) in urine samples of Italian women residing in the north, centre, and south of Italy. For BPA data, rural and urban areas are also considered. Data were normalized to urine creatinine concentration (µg/g). Two outlier values were excluded from the graph for a better visualization. Black lines and asterisks indicate statistical differences among macro-areas; red and green bars/asterisks indicate statistical differences between rural and urban areas in the centre and south of Italy, respectively. Asterisks indicate the level of significance: * *p* < 0.05, ** *p* < 0.01, and *** *p* < 0.001.

**Table 1 ijms-23-16012-t001:** Distribution of enrolled women age and BMI in Italian macro-areas and areas. Median values and interquartile range are indicated; *p*-values for significant differences among groups are indicated in the beside column.

	N	Age	*p*-Value	N	BMI	*p*-Value
		P50 (P25–P75)			P50 (P25–P75)	
TOT	655	41 (38–44)		879	22.3 (20.3–24.8)	
North	254	41 (38–44)		295	21.8 (20.0–24.1)	0.0062 ^N vs. S^
Centre	193	41 (38–44)		290	22.5 (20.2–24.6)	0.0348 ^C vs. N^
South	208	40 (36–44)	0.0390 ^S vs. C^	293	22.8 (21.1–26.0)	<0.0001 ^S vs. N^
Rural	319	40 (36–43)		439	22.3 (20.6–25.0)	
Urban	336	41 (38–45)	0.0024 ^U vs. R^	440	22.3 (20.3–24.7)	

Different superscript letters indicate statistically significant differences between groups (N, north; C, centre; S, south; R, rural; U, urban).

**Table 2 ijms-23-16012-t002:** DEHP metabolite and BPA levels in urine samples of 898 Italian women. In the table are reported the percentage of the samples above the limit of detection (LOD), the geometric mean (GM) with the 95% confidence interval (CI), and the median (P50) with the interquartile range (P25–P75). For each metabolite, the first row refers to the unadjusted concentration in urine (μg/L) and the second row to the creatinine-adjusted concentration (μg/g).

Metabolite	Unit	>LOD (%)	GM (95% CI)	P50 (P25–P75)
**MEHP**	µg/L	99.33%	4.64 (4.39–4.91)	5.10 (3.37–7.41)
	µg/g crea	99.33%	4.30 (4.04–4.58)	4.50 (2.92–6.84)
**MEHHP**	µg/L	97.44%	18.0 (17.0–19.1)	18.0 (10.5–31.7)
	µg/g crea	97.44%	16.6 (15.7–17.5)	16.1 (10.2–26.4)
**MEOHP**	µg/L	98.87%	5.53 (5.21–5.87)	5.62 (3.20–9.64)
	µg/g crea	98.87%	5.11 (4.84–5.38)	4.91 (3.14–7.98)
**Σ PHTHALATES**	µg/L	99.33%	29.7 (28.2–31.3)	29.5 (18.1–47.6)
	µg/g crea	99.33%	27.5 (26.3–28.8)	26.6 (17.6–41.2)
**BPA**	µg/L	95.65%	5.79 (5.41–6.20)	5.76 (3.30–10.3)
	µg/g crea	95.65%	5.36 (4.98–5.77)	5.06 (3.14–9.73)

**Table 3 ijms-23-16012-t003:** Levels of the sum of DEHP metabolites in urine samples of women residing in the north (N = 300), centre (N = 299), or south (N = 299) of Italy. In the table are reported the median (P50) with the interquartile (P25–P75) range for both unadjusted (µg/L) and creatinine-adjusted concentrations (µg/g).

		ΣDEPH Metabolites		BPA	
	Unit	P50 (P25–P75)	*p*-Value	P50 (P25–P75)	*p*-Value
**MACRO-AREA**					
NORTH	µg/L	25.6 (15.8–41.8) *	<0.0001 ^N vs. S^	6.88 (3.98–14.9) *	
CENTRE		26.9 (17.3–45.8) *	0.0001 ^C vs. S^	5.62 (3.25–8.58) *	0.0001 ^C vs. N^
SOUTH		35.8 (21.6–56.8) *		5.25 (3.05–8.72) *	<0.0001 ^S vs. N^
NORTH	µg/g crea	25.3 (17.5–38.8) *	0.0007 ^N vs. S^	6.37 (3.78–14.9) *	
CENTRE		24.4 (15.9–38.00) *	0.0001 ^C vs. S^	4.75 (3.17–8.26) *	<0.0001 ^C vs. N^
SOUTH		28.9 (20.3–46.5) *		4.72 (2.76–7.73 *	<0.0001 ^S vs. N^
**AREA**					
RURAL	µg/L	29.0 (18.3–46.9)		5.44 (3.09–9.78) *	
URBAN		29.9 (17.8–49.9)		6.04 (3.76–10.7) *	0.0181 ^U vs. R^
RURAL	µg/g crea	25.7 (17.5–39.4)		4.76 (2.80–8.26) *	
URBAN		27.3 (18.1–41.7)		5.41 (3.54–10.5) *	0.0009 ^U vs. R^

* Statistically significant differences among macro-area and area. Different superscript letters indicate statistically significant differences between groups (N, north; C, centre; S, south; R, rural; U, urban).

**Table 4 ijms-23-16012-t004:** Levels of the sum of DEHP metabolites and BPA in urine samples of women residing in rural or urban Italian areas in each macro-area. In the table are reported the median (P50) with the interquartile range (P25–P75) for both unadjusted (ng/mL) and creatinine-adjusted concentrations (μg/g).

				ΣDEPH Metabolites			BPA		
Macro-Area	Area	N	Unit	P50 (P25–P75)	*p*-Value (Area)	*p*-Value (MA)	P50 (P25–P75)	*p*-Value (Area)	*p*-Value (MA)
**NORTH**	Rural	150	µg/L	25.3 (15.9–41.3)			7.41 (4.09–17.9)		
	Urban	150		26.5 (15.8–42.2)			6.26 (3.89–12.4)		
	Rural	150	µg/g crea	25.0 (17.2–37.7)			6.28 (3.47–18.8)		
	Urban	150		25.9 (18.0–40.9)			6.53 (3.99–12.5)		
**CENTRE**	Rural	149	µg/L	24.9 (16.6–45.8)			5.29 (3.01–7.26)		<0.0001 ^C vs. N^
	Urban	150		28.8 (17.8–46.1)			6.09 (3.65–9.49)	0.0019 ^U vs. R^	
	Rural	149	µg/g crea	22.9 (14.9–34.6)			4.39 (2.60–7.12)		<0.0001 ^C vs. N^
	Urban	150		27.5 (18.0–38.8)	0.0401 ^U vs. R^		5.12 (3.56–10.3)	0.0006 ^U vs. R^	
**SOUTH**	Rural	149	µg/L	35.0 (24.5–52.3)		0.0003 ^S vs. N^	4.33 (2.63–8.10)		<0.0001 ^S vs. N^
						0.0003 ^S vs. C^			
	Urban	150		37.0 (18.9–57.6)		0.0018 ^S vs. N^	5.86 (3.69–9.58)	0.0092 ^U vs. R^	
						0.0162 ^S vs. C^			
	Rural	149	µg/g crea	29.9 (21.9–43.4)		0.0029 ^S vs. N^	4.02 (2.31–7.30)		<0.0001 ^S vs. N^
						<0.0001 ^S vs. C^			
	Urban	150		28.0 (18.5–47.6)		0.0385 ^S vs. C^	5.16 (3.07–8.65)	0.0073 ^U vs. R^	0.0047 ^S vs. N^

Different superscript letters indicate statistically significant differences between groups (N, north; C, centre; S, south; R, rural; U, urban).

**Table 5 ijms-23-16012-t005:** Relative metabolic rates and percentage fractions of DEHP metabolites in women residing in the three macro-areas and in rural vs urban areas. Data are expressed as medians (IQ range).

AREA		P50 (P25–P75)	*p*-Value
Macro-Area			
**NORTH**	RMR1	4.69 (2.79–7.40)	0.0046 ^N vs. S^
	RMR2	3.18 (2.44–4.01)	0.0053 ^N vs. C^
	%MEHP	17.6 (11.9–26.4)	0.0046 ^N vs. S^
	%MEHHP	62.3 (43.3–68.3)	
	%MEOHP	19.2 (15.5–22.7)	0.0085 ^N vs. C^
**CENTRE**	RMR1	4.56 (2.71–6.97)	0.0017 ^C vs. S^
	RMR2	2.78 (2.06–4.01)	<0.0001 ^C vs. S^
	%MEHP	18.0 (12.6–27.0)	0.0017 ^C vs. S^
	%MEHHP	63.3 (54.6–70.1)	
	%MEOHP	17.2 (13.8–22.7)	<0.0001 ^C vs. S^
**SOUTH**	RMR1	5.51 (3.44–8.30)	
	RMR2	3.37 (2.56–4.84)	0.0187 ^S vs. N^
	%MEHP	15.4 (10.8–22.5)	
	%MEHHP	62.1 (53.8–70.7)	
	%MEOHP	20.9 (17.1–26.0)	0.0001 ^S vs. N^
Area			
**RURAL**	RMR1	4.82 (2.83–7.62)	
	RMR2	3.13 (2.42–4.12)	
	%MEHP	17.2 (11.6–26.1)	
	%MEHHP	62.1 (53.0–69.0)	
	%MEOHP	19.4 (15.3–23.2)	
**URBAN**	RMR1	4.97 (3.15–7.55)	
	RMR2	3.15 (2.22–4.32)	
	%MEHP	16.7 (11.7–24.1)	
	%MEHHP	62.7 (54.4–70.3)	
	%MEOHP	19.2 (15.0–24.0)	

Different superscript letters indicate statistically significant differences between groups (N, north; C, centre; S, south).

**Table 6 ijms-23-16012-t006:** Estimated daily intake (µg/kg bw per day) of BPA in Italian women (N = 879) according to residing areas and age.

Category	Sub-Category (a)	Sub-Category (b)	N	Daily Intake GM (95% CI) (µg/kg bw Day)	*p*-Value (a)	*p*-Value (b)
Total			879	0.11 (0.11–0.12)		
Macro-area	North		295	0.15 (0.13–0.17)		
	Centre		290	0.098 (0.086–0.11)	0.0001 ^C vs. N^	
	South		294	0.10 (0.092–0.11)	<0.0001 ^S vs. N^	
Area	Rural		439	0.10 (0.093–0.12)		
	Urban		440	0.13 (0.12–0.14)	0.0123 ^U vs. R^	
Age class	20–30 y		19	0.099 (0.061–0.16)		
	30–40 y ^a^		302	0.12 (0.11–0.14)	0.0070 ^a vs. b^	
	40–50 y ^a^		329	0.11 (0.10–0.13)	0.0115 ^a vs. b^	
	>50 y ^b^		14	0.050 (0.024–0.11)		
Macro-area/Area	North	Rural	146	0.16 (0.13–0.20)		
		Urban	149	0.14 (0.12–0.16)		
	Centre	Rural	145	0.074 (0.060–0.091)	<0.0001 ^C vs. N^	
		Urban	145	0.13 (0.11–0.15)		0.0035 ^U vs. R^
	South	Rural	148	0.094 (0.081–0.11)	<0.0001 ^S vs. N^	
		Urban	146	0.11 (0.098–0.13)	0.0165 ^S vs. N^	0.0197 ^U vs. R^

Different superscript letters indicate statistically significant differences between groups (a, b; N, north; C, centre; S, south; R, rural; U, urban).

**Table 7 ijms-23-16012-t007:** Estimated daily intake (µg/kg bw per day) of DEHP in Italian women (N = 881) according to residing areas and age.

Category	Sub-Category (a)	Sub-Category (b)	N	Daily Intake GM (95% CI) (µg/kg bw Day)	*p*-Value (a)
Total			881	4.86 (4.66–5.08)	
Macro-area	North		296	4.64 (4.33–4.97)	
	Centre		290	4.56 (4.22–4.93)	0.0002 ^C vs. S^
	South		295	5.44 (5.02–5.88)	0.0017 ^S vs. N^
Area	Rural		436	4.79 (4.50–5.09)	
	Urban		445	4.94 (4.65–5.26)	
Age class	20–30 y		19	4.31 (3.15–5.89)	
	30–40 y		302	4.57 (4.26–4.89)	
	40–50 y		332	4.94 (4.61–5.28)	
	>50 y		15	5.09 (3.55–7.30)	
Macro-area/Area	North	Rural	147	4.53 (4.08–5.03)	0.0008 ^N vs. S^
		Urban	149	4.75 (4.33–5.21)	
	Centre	Rural	143	4.39 (3.90–4.94)	0.0001 ^C vs. S^
		Urban	147	4.74 (4.26–5.26)	
	South	Rural	146	5.51 (5.00–6.08)	
		Urban	149	5.36 (4.74–6.07)	0.0440 ^S vs. N^

Different superscript letters indicate statistically significant differences between groups (N, north; C, centre; S, south.

**Table 8 ijms-23-16012-t008:** Reference values (RV95, P95(95% CI)) for the sum of DEHP metabolites and BPA in urine of women according to the residing area/macro-area and to age categories.

Category	Sub-Category (a)	Sub-Category (b)	N	Σ DEHP RV95 [µg/L] (95% CI)	N	BPA RV95 [µg/L] (95% CI)
Total			892	101 (93.6–117)	898	30.6 (25.5–37.5)
Macro-area	North		300	96.4 (80.8–111)	300	54.6 (38.6–66.8)
	Centre		295	94.3 (81.1–141)	299	23.9 (19.7–31.5)
	South		297	124 (99.8–197)	299	20.4 (16.6–25.2)
Area	Rural		446	98.5 (87.2–139)	448	39.4 (26.2–51.4)
	Urban		446	107 (94.7–125)	450	27.2 (22.5–32.9)
Age class	20–30 y		19	188 (50.5–188)	19	25.3 (13.4–25.3)
	30–40 y		303	99.5 (85.0–123)	304	30.4 (24.5–39.9)
	40–50 y		333	101 (88.6–125)	334	25.4 (21.7–38.7)
	>50 y		15	57.3 (45.6–57.3)	15	15.0 (7.3–15.0)
Macro-area/Area	North	Rural	150	96.2 (71.7–139)	150	70.0 (51.0–94.0)
		Urban	150	97.9 (80.4–112)	150	35.2 (24.5–52.0)
	Centre	Rural	148	94.9 (73.9–327)	149	19.5 (13.4–26.4)
		Urban	147	94.5 (71.8–130.8)	150	29.7 (21.0–49.4)
	South	Rural	148	113.4 (88.7–211)	149	22.8 (15.8–38.6)
		Urban	149	152 (106–308)	150	19.6 (14.8–26.2)

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
