# Peer review of "Exposure to Endocrine Disruptors (Di(2-Ethylhexyl)phthalate (DEHP) and Bisphenol A (BPA)) in Women from Different Residing Areas in Italy: Data from the LIFE PERSUADED Project"

_ijms, 2022, doi:10.3390/ijms232416012_

Round 1
Reviewer 1 Report
The article by Carli et al. “Exposure to endocrine disruptors (Di(2-ethylhexyl)phthalate (DEHP) and bisphenol A (BPA)) in women from different residing areas in Italy: data from the LIFE PERSUADED project” reports results from the large biomonitoring study LIFE PERSUADED. This project was funded by the European Commission and was aimed to define the co-exposure in Italian mother-child pairs to DEHP metabolites and BPA, two well-known endocrine disrupters. 900 mother-child pairs were enrolled for 3 years (2015-17) from urban and rural AREAS located in 3 MACROAREAS: North, Centre and South Italy. Levels of DEHP metabolites and BPA were measured in urine samples by high sensitivity mass spectrometry and exposure levels are also reported according to residing areas. The same authors previously published the results on the children, the present work reports on the mothers.
The study has been well designed, is methodologically accurate and the results are highly relevant for determination of risk assessment. The large number of mother-child pairs enrolled and the amount of data collected during the project provides reliable calculation of daily intake and reference values for both DEHP and BPA in the Italian female population, including relation to the geographical area. Therefore I highly recommend publication of this manuscript.
I have only minor observations and suggestions, regarding mainly text errors:
- Lines 98-107: please remove the whole paragraph from “introduction” to “references” regarding authors’ instructions to write the Introduction.
- Lines 116-119: the sentence from ”this paper” to “[22, 23]” was already reported at the end of the Introduction (lines 95-97). Please remove it.
- Lines 211- 220: the paragraph from “Clinical” to “[22, 23]” was already reported at the beginning of the Result section (lines 110-119). Please remove it.
- Lines 272-276: the paragraph from “calculated” to “per day” was already reported at the beginning of section 2.5 (lines 240-244). Please remove it.
- Lines 284-288: the paragraph from “Sum” to “areas” was already reported at the beginning of section 2.6 (lines 264-267). Please remove it.
- Lines 313-314: this sentence is probably missing some words and is not clear
- Lines 357-358: “we can”?
- All references from line 525 are numbered twice.
Author Response
We thank the reviewer for the comments that allowed us to improve the manuscript. We have revised the entire manuscript also adding a new table with the report of Daily Intake for DEHP.
As suggested by the reviewer sentences (Lines 98-107; Lines 116-119; Lines 211- 220; Lines 284-288) were deleted and the sentences 313-314 and 357-358 were corrected. The double reference number from line 525 was corrected. Track change was used to show in the text the modifications compared to original manuscript.
Reviewer 2 Report
line 172: MEHP (ug/L or ug/g crea)
lines 355-357: of 4 ug/Kg per day established by EFSA in 2015 [4], while the DEHP daily intake was assumed below the EFSA limit of 50 ug/kg per day given that the sum of DEHP metabolites was comparable to European data [28] we can. BPA daily intake in Italian women was 0.11 ug/kg per day,
"ug" must change "µg".
Author Response
We thank the reviewer for the comments that allowed us to improve the manuscript. We have revised the entire manuscript also adding a new table with the report of Daily Intake for DEHP as suggested.
Track change was used to show in the text the modifications compared to original manuscript.
Reviewer 3 Report
General comments
In principle, the manuscript reports interesting results from relevant research. However, it seems to me that something went wrong. Did you provide, perhaps, an unfinished draft instead of a final article ready for review? My observations leading to this conclusion were the following:
In the introduction, the lines 98-107, starting with "introduction", are superfluous, not directly related to your research and apparently erroneously copied from the journal's guidance. Must be deleted.
It is surprising and unusual to read "LIFE PERSUADED Project Group" in the author's list. I would suggest to delete it and, instead, to explain briefly in an appropriate place what this project is. (I wonder if the information given in the introduction on line 88 is sufficient or must be extended. Personally, I would suggest the latter.)
In the "Results" section, the information given on lines 110 to 119 is exactly the same as on lines 211 to 220. and, thus, redundant. This can't be intended but must be an error.
In the reference list, all figures are given twice, e.g., on line 618: 32. 32.Baralic, K.; ... The "double numbering" must be corrected.
There might be more examples throughout the text that is certainly in an urgent need to be checked. In my understanding, this task is not a reviewer's responsibility. Accordingly, I don't see another chance than to suggest rejection of the manuscript and to recommend complete revision before a new submission.
Specific comments
Abstract, introduction, discussion:
A reader might wonder if there are any TDIs (perhaps proposed) for DEHP. If this is not the case, in contrast to BPA, it should be clearly stated.
Results
Lines 202-206: Looking at the table below, in particular the quartiles, I seriously doubt the assumption of a higher metabolic conversion rate of MEHP in women from Southern Italy. Seems to me a very brave hypothesis. Indeed, it is discussed then in length below. The "Discussion" is the right place for that (even though I still feel that is given too much weight) but in the "Results" section I would just report the findings without interpretation.
2.5 Daily intake of BPA: Even though the title of this section, it is about BPA until line 250 only Then, it is, again, about DEHP metabolites and reference is made, once more, to Table 5. Must be thoroughly checked.
Discussion
The sentence on lines 313 and 314 is not complete. I can hardly guess what is meant. Most likely, it is about exposure.
The assumption of differences in metabolism appears a bit speculative but, of course, it is up to the authors to put forward this hypothesis.
Author Response
Response to reviewer:
We thank the reviewer for the comments that allowed us to improve the manuscript. Indeed, there was a problem when the paper was formatted for IJMS that requires a specific template rather than a simple word file with tables and figures at the end of the manuscript.
Please find below the point by point response to the specific comments. Track change was used to show in the text the modifications compared to original manuscript
- In the introduction, the lines 98-107, starting with "introduction", are superfluous, not directly related to your research and apparently erroneously copied from the journal's guidance. Must be deleted.
ANSWER: Thanks for pointing this out. These lines were in the template and by mistake were not removed before submission. The sentences are now deleted
- It is surprising and unusual to read "LIFE PERSUADED Project Group" in the author's list. I would suggest to delete it and, instead, to explain briefly in an appropriate place what this project is. (I wonder if the information given in the introduction on line 88 is sufficient or must be extended. Personally, I would suggest the latter.)
ANSWER: As in previous papers we acknowledged among the authors the consortium whose names are listed at the end of the manuscript. This is mentioned now in the notes to the list of authors
- In the "Results" section, the information given on lines 110 to 119 is exactly the same as on lines 211 to 220. and, thus, redundant. This can't be intended but must be an error.
In the reference list, all figures are given twice, e.g., on line 618: 32. 32.Baralic, K.; ... The "double numbering" must be corrected.
There might be more examples throughout the text that is certainly in an urgent need to be checked. In my understanding, this task is not a reviewer's responsibility. Accordingly, I don't see another chance than to suggest rejection of the manuscript and to recommend complete revision before a new submission.
ANSWER: Thanks for pointing this out. We apologize for these errors that have been corrected. We have checked and completely revised the test. Extra lines are now deleted and reference list has been updated after changes in the text
- Abstract, introduction, discussion: A reader might wonder if there are any TDIs (perhaps proposed) for DEHP. If this is not the case, in contrast to BPA, it should be clearly stated.
ANSWER: The Daily Intake for DEHP has been calculated and inserted in the text and discussed. TABLE 7 was added with the values of DI for DEHP.
- Results Lines 202-206: Looking at the table below, in particular the quartiles, I seriously doubt the assumption of a higher metabolic conversion rate of MEHP in women from Southern Italy. Seems to me a very brave hypothesis. Indeed, it is discussed then in length below. The "Discussion" is the right place for that (even though I still feel that is given too much weight) but in the "Results" section I would just report the findings without interpretation.
Discussion The assumption of differences in metabolism appears a bit speculative but, of course, it is up to the authors to put forward this hypothesis.
ANSWER: relative metabolic rates were not assumed but calculated according to Kim et al PLoS One 2018,
RMR1, representing the metabolic conversion of MEHP into the secondary metabolites (MEHHP+MEOHP), was 5.51 in South vs 4.69 in North and 4.56 in Centre Italy, all p<0.009. RMR2, representing the rate of oxidation from MEHHP to MEOHP, was 3.37 in South vs 3.18 in North and 2.78 in Centre Italy, all p<0.006
Results 2.5 Daily intake of BPA: Even though the title of this section, it is about BPA until line 250 only Then, it is, again, about DEHP metabolites and reference is made, once more, to Table 5. Must be thoroughly checked.
ANSWER: We have revised the entire paragraph that now also includes the report of DI for DEHP
Discussion The sentence on lines 313 and 314 is not complete. I can hardly guess what is meant. Most likely, it is about exposure.
Round 2
Reviewer 3 Report
The manuscript has been considerably improved meanwhile and is now in state that it could be published. However, I have noticed some further points that might warrant your consideration. Therefore, I have compiled very few additional specific comments.
Abstract
Lines 40/41: I am not sure about the meaning of "contrasting the exposure". Do you suggest to reduce the exposure (and, this way, the risk of adverse health outcomes)? This is likely becaue of line 352 in the "Discussion". (But also there, "contrasting" does not appear the best wording.) Or do you recommend monitoring of BPA and DEHP metabolites levels? Or both? Please clarify!
Introduction
Line 84: Is there a separate reference on re-evaluation of BPA by the FDA? (See also line 345 in the "Discussion".) Actually, I could only find information on EFSA's activities.
Lines 105/106: The last two sentences of the introduction, beginning with "Introduction" are, apparently, still leftovers from the journal's template. (I wonder how they could be kept there because you got succesfully rid of the others...)
Author Response
Lines 40/41: I am not sure about the meaning of "contrasting the exposure". Do you suggest to reduce the exposure (and, this way, the risk of adverse health outcomes)? This is likely because of line 352 in the "Discussion". (But also there, "contrasting" does not appear the best wording.) Or do you recommend monitoring of BPA and DEHP metabolites levels? Or both? Please clarify!
ANSWER: We agree that the word contrasting is not appropriate and we have changed it as suggested.
Line 84: Is there a separate reference on re-evaluation of BPA by the FDA? (See also line 345 in the "Discussion".) Actually, I could only find information on EFSA's activities.
ANSWER: We have updated the text with the info regarding the filed petition
Other info can be found at this link:
https://www.foodpackagingforum.org/news/us-fda-to-review-safety-of-bpa-in-food-contact
Lines 105/106: The last two sentences of the introduction, beginning with "Introduction" are, apparently, still leftovers from the journal's template. (I wonder how they could be kept there because you got succesfully rid of the others...)
ANSWER: Sorry, we also thought we have deleted all the leftovers